# Multi-Agent Generative Adversarial Imitation Learning

**Jiaming Song**
Stanford University
tsong@cs.stanford.edu

**Hongyu Ren**
Stanford University
hyren@cs.stanford.edu

**Dorsa Sadigh**
Stanford University
dorsa@cs.stanford.edu

**Stefano Ermon**
Stanford University
ermon@cs.stanford.edu

## Abstract

Imitation learning algorithms can be used to learn a policy from expert demonstrations without access to a reward signal. However, most existing approaches are not applicable in multi-agent settings due to the existence of multiple (Nash) equilibria and non-stationary environments. We propose a new framework for multi-agent imitation learning for general Markov games, where we build upon a generalized notion of inverse reinforcement learning. We further introduce a practical multi-agent actor-critic algorithm with good empirical performance. Our method can be used to imitate complex behaviors in high-dimensional environments with multiple cooperative or competing agents.

## 1 Introduction

Reinforcement learning (RL) methods are becoming increasingly successful at optimizing reward signals in complex, high dimensional environments [1]. A key limitation of RL, however, is the difficulty of designing suitable reward functions for complex and not well-specified tasks [2, 3]. If the reward function does not cover all important aspects of the task, the agent could easily learn undesirable behaviors [4]. This problem is further exacerbated in multi-agent scenarios, such as multiplayer games [5], multi-robot control [6] and social interactions [7]; in these cases, agents do not even necessarily share the same reward function and might even have conflicting rewards.

Imitation learning methods address these problems via expert demonstrations [8–11]; the agent directly learns desirable behaviors by imitating an expert. Notably, inverse reinforcement learning (IRL) frameworks assume that the expert is (approximately) optimizing an underlying reward function, and attempt to recover a reward function that rationalizes the demonstrations; an agent policy is subsequently learned through RL [12, 13]. Unfortunately, this paradigm is not suitable for general multi-agent settings due to environment being non-stationary to individual agents [14] and the existence of multiple equilibrium solutions [15]. The optimal policy of one agent could depend on the policies of other agents, and vice versa, so there could exist multiple solutions in which each agents' policy is the optimal response to others.

In this paper, we propose a new framework for multi-agent imitation learning – provided with demonstrations of a set of experts interacting with each other in the same environment, we aim to learn multiple parametrized policies that imitate the behavior of each expert respectively. Using the framework of Markov games, we integrate multi-agent RL with a suitable extension of multi-agent inverse RL. The resulting procedure strictly generalizes Generative Adversarial Imitation Learning (GAIL, [16]) in the single agent case. Imitation learning in our setting corresponds to a two-player

game between a generator and a discriminator. The generator controls the policies of all the agents in a distributed way, and the discriminator contains a classifier for each agent that is trained to distinguish that agent's behavior from that of the corresponding expert. Upon training, the behaviors produced by the policies should be indistinguishable from the training data. We can incorporate prior knowledge into the discriminators, including the presence of cooperative or competitive agents. In addition, we propose a novel multi-agent natural policy gradient algorithm that addresses the issue of high variance gradient estimates commonly observed in reinforcement learning [14, 17]. Empirical results demonstrate that our method can imitate complex behaviors in high-dimensional environments, such as particle environments and cooperative robotic control tasks, with multiple cooperative or competitive agents; the imitated behaviors are close to the expert behaviors with respect to "true" reward functions which the agents do not have access to during training.

## 2 Preliminaries

### 2.1 Markov games

We consider an extension of Markov decision processes (MDPs) called Markov games [18]. A Markov game (MG) for $N$ agents is defined via a set of states $\mathcal{S}$, and $N$ sets of actions $\{\mathcal{A}_i\}_{i=1}^N$. The function $P : \mathcal{S} \times \mathcal{A}_1 \times \cdots \times \mathcal{A}_N \to \mathcal{P}(\mathcal{S})$ describes the (stochastic) transition process between states, where $\mathcal{P}(\mathcal{S})$ denotes the set of probability distributions over the set $\mathcal{S}$. Given that we are in state $s_t$ at time $t$, the agents take actions $(a_1, \ldots, a_N)$ and the state transitions to $s_{t+1}$ with probability $P(s_{t+1}|s_t, a_1, \ldots, a_N)$. Each agent $i$ obtains a (bounded) reward given by a function $r_i : \mathcal{S} \times \mathcal{A}_1 \times \cdots \times \mathcal{A}_N \to \mathbb{R}$. Each agent $i$ aims to maximize its own total expected return $R_i = \sum_{t=0}^\infty \gamma^t r_{i,t}$, where $\gamma$ is the discount factor, by selecting actions through a (stationary and Markovian) stochastic policy $\pi_i : \mathcal{S} \times \mathcal{A}_i \to [0, 1]$. The initial states are determined by a distribution $\eta : \mathcal{S} \to [0, 1]$. The joint policy is defined as $\boldsymbol{\pi}(\mathbf{a}|s) = \prod_{i=1}^N \pi_i(a_i|s)$, where we use bold variables without subscript $i$ to denote the concatenation of all variables for all agents (e.g., $\boldsymbol{\pi}$ denotes the joint policy $\prod_{i=1}^N \pi_i$ in a multi-agent setting, $\mathbf{r}$ denotes all rewards, $\mathbf{a}$ denotes actions of all agents). We use expectation with respect to a policy $\pi$ to denote an expectation with respect to the trajectories it generates, and use subscript $-i$ to denote *all agents except for $i$*. For example, $(a_i, a_{-i})$ represents $(a_1, \ldots, a_N)$, the actions of all $N$ agents.

### 2.2 Reinforcement learning and Nash equilibrium

In reinforcement learning (RL), the goal of each agent is to maximize total expected return $\mathbb{E}_\pi[r(s, a)]$ given access to the reward signal $r$. In single agent RL, an optimal Markovian policy exists but the optimal policy might not be unique (e.g., all policies are optimal for an identically zero reward; see [19], Chapter 3.8). An entropy regularizer can be introduced to resolve this ambiguity. The optimal policy is found via the following RL procedure:

$$\text{RL}(r) = \underset{\pi \in \Pi}{\arg\max} \, H(\pi) + \mathbb{E}_\pi[r(s, a)], \tag{1}$$

where $H(\pi)$ is the $\gamma$-discounted causal entropy [20] of policy $\pi \in \Pi$.

**Definition 1** ($\gamma$-discounted Causal Entropy). *The $\gamma$-discounted causal entropy for a policy $\pi$ is defined as follows:*

$$H(\pi) \triangleq \mathbb{E}_\pi[-\log \pi(a|s)] = \mathbb{E}_{s_t, a_t \sim \pi}\left[-\sum_{t=0}^\infty \gamma^t \log \pi(a_t|s_t)\right].$$

The addition of $H(\pi)$ in (1) resolves this ambiguity – the policy with both the highest reward and the highest entropy[1] is unique because the entropy function is strictly concave with respect to $\pi$.

In Markov games, however, the optimal policy of an agent depends on other agents' policies. One approach is to use an equilibrium solution concept, such as Nash equilibrium [15]. Informally, a set of policies $\{\pi_i\}_{i=1}^N$ is a Nash equilibrium if no agent can achieve higher reward by unilaterally

changing its policy, i.e., $\forall i \in [1, N], \forall \hat{\pi}_i \neq \pi_i, \mathbb{E}_{\pi_i, \pi_{-i}}[r_i] \geq \mathbb{E}_{\hat{\pi}_i, \pi_{-i}}[r_i]$. The process of finding a Nash equilibrium can be defined as a constrained optimization problem ([21], Theorem 3.7.2):

$$\min_{\boldsymbol{\pi} \in \Pi, \mathbf{v} \in \mathbb{R}^{\mathcal{S} \times N}} f_r(\boldsymbol{\pi}, \mathbf{v}) = \sum_{i=1}^{N} \left( \sum_{s \in \mathcal{S}} v_i(s) - \mathbb{E}_{a_i \sim \pi_i(\cdot|s)} q_i(s, a_i) \right) \tag{2}$$

$$v_i(s) \geq q_i(s, a_i) \triangleq \mathbb{E}_{\pi_{-i}} \left[ r_i(s, \mathbf{a}) + \gamma \sum_{s' \in \mathcal{S}} P(s'|s, \mathbf{a}) v_i(s') \right] \quad \forall i \in [N], s \in \mathcal{S}, a_i \in \mathcal{A}_i \tag{3}$$

$$\mathbf{a} \triangleq (a_i, a_{-i}) \triangleq (a_1, \dots, a_N) \qquad \mathbf{v} \triangleq [v_1; \dots; v_N]$$

where the joint action $\mathbf{a}$ includes actions $a_{-i}$ sampled from $\pi_{-i}$ and $a_i$. Intuitively, $\mathbf{v}$ can be thought of as a value function and $\mathbf{q}$ represents the $Q$-function that corresponds to $\mathbf{v}$. The constraints enforce the Nash equilibrium condition – when the constraints are satisfied, $(v_i(s) - q_i(s, a_i))$ is non-negative for every $i \in [N]$. Hence $f_r(\boldsymbol{\pi}, \mathbf{v})$ is always non-negative for a feasible $(\boldsymbol{\pi}, \mathbf{v})$. Moreover, this objective has a global minimum of zero if a Nash equilibrium exists, and $\pi$ forms a Nash equilibrium if and only if $f_r(\boldsymbol{\pi}, \mathbf{v})$ reaches zero while being a feasible solution ([22], Theorem 2.4).

## 2.3 Inverse reinforcement learning

Suppose we do not have access to the reward signal $r$, but have demonstrations $\mathcal{D}$ provided by an expert ($N$ expert agents in Markov games). Imitation learning aims to learn policies that behave similarly to these demonstrations. In Markov games, we assume all experts/players operate in the same environment, and the demonstrations $\mathcal{D} = \{(s_j, \mathbf{a}_j)\}_{j=1}^{M}$ are collected by sampling $s_0 \sim \eta(s), \mathbf{a}_t = \pi_E(\mathbf{a}_t|s_t), s_{t+1} \sim P(s_{t+1}|s_t, \mathbf{a}_t)$; we assume knowledge of $N, \gamma, \mathcal{S}, \mathcal{A}$, as well as access to $T$ and $\eta$ as black boxes. We further assume that once we obtain $\mathcal{D}$, we cannot ask for additional expert interactions with the environment (unlike in DAgger [23] or CIRL [24]).

Let us first consider imitation in Markov decision processes (as a special case to Markov games) and the framework of single-agent Maximum Entropy IRL [8, 16] where the goal is to recover a reward function $r$ that rationalizes the expert behavior $\pi_E$:

$$\text{IRL}(\pi_E) = \arg\max_{r \in \mathbb{R}^{\mathcal{S} \times \mathcal{A}}} \mathbb{E}_{\pi_E}[r(s, a)] - \left( \max_{\pi \in \Pi} H(\pi) + \mathbb{E}_{\pi}[r(s, a)] \right)$$

In practice, expectations with respect to $\pi_E$ are evaluated using samples from $\mathcal{D}$.

The IRL objective is ill-defined [12, 10] and there are often multiple valid solutions to the problem when we consider all $r \in \mathbb{R}^{\mathcal{S} \times \mathcal{A}}$. To resolve this ambiguity, [16] introduce a convex reward function regularizer $\psi : \mathbb{R}^{\mathcal{S} \times \mathcal{A}} \to \mathbb{R}$, which can be used for example to restrict rewards to be linear in a pre-determined set of features [16]:

$$\text{IRL}_\psi(\pi_E) = \arg\max_{r \in \mathbb{R}^{\mathcal{S} \times \mathcal{A}}} -\psi(r) + \mathbb{E}_{\pi_E}[r(s, a)] - \left( \max_{\pi \in \Pi} H(\pi) + \mathbb{E}_{\pi}[r(s, a)] \right) \tag{4}$$

## 2.4 Imitation by matching occupancy measures

[16] interprets the imitation learning problem as matching two occupancy measures, i.e., the distribution over states and actions encountered when navigating the environment with a policy. Formally, for a policy $\pi$, it is defined as $\rho_\pi(s, a) = \pi(a|s) \sum_{t=0}^{\infty} \gamma^t P(s_t = s|\pi)$. [16] draws a connection between IRL and occupancy measure matching, showing that the former is a dual of the latter:

**Proposition 1** (Proposition 3.1 in [16])**.**

$$\text{RL} \circ \text{IRL}_\psi(\pi_E) = \arg\min_{\pi \in \Pi} -H(\pi) + \psi^\star(\rho_\pi - \rho_{\pi_E})$$

Here $\psi^\star(x) = \sup_y x^\top y - \psi(y)$ is the convex conjugate of $\psi$, which could be interpreted as a measure of similarity between the occupancy measures of expert policy and agent's policy. One instance of $\psi = \psi_{\text{GA}}$ gives rise to the Generative Adversarial Imitation Learning (GAIL) method:

$$\psi_{\text{GA}}^\star(\rho_\pi - \rho_{\pi_E}) = \max_{D \in (0,1)^{\mathcal{S} \times \mathcal{A}}} \mathbb{E}_{\pi_E}[\log(D(s, a))] + \mathbb{E}_\pi[\log(1 - D(s, a))] \tag{5}$$

The resulting imitation learning method from Proposition 1 involves a *discriminator* (a classifier $D$) competing with a *generator* (a policy $\pi$). The discriminator attempts to distinguish real vs. synthetic trajectories (produced by $\pi$) by optimizing (5). The generator, on the other hand, aims to perform optimally under the reward function defined by the discriminator, thus "fooling" the discriminator with synthetic trajectories that are difficult to distinguish from the expert ones.

## 3 Generalizing IRL to Markov games

Extending imitation learning to multi-agent settings is difficult because there are multiple rewards (one for each agent) and the notion of optimality is complicated by the need to consider an equilibrium solution [15]. We use MARL($\mathbf{r}$) to denote the set of (stationary and Markovian) policies that form a Nash equilibrium under $r$ and have the maximum $\gamma$-discounted causal entropy (among all equilibria):

$$\text{MARL}(\mathbf{r}) = \underset{\boldsymbol{\pi} \in \Pi, \mathbf{v} \in \mathbb{R}^{S \times N}}{\arg\min} f_r(\boldsymbol{\pi}, \mathbf{v}) - H(\boldsymbol{\pi}) \tag{6}$$

$$v_i(s) \geq q_i(s, a_i) \quad \forall i \in [N], s \in \mathcal{S}, a_i \in \mathcal{A}_i$$

where $q$ is defined as in Equation (3). Our goal is to define a suitable inverse operator MAIRL, in analogy to IRL in Equation (4), which chooses a reward that creates a *margin* between the expert and every other policy. However, the *constraints* in the Nash equilibrium optimization (Equation (6)) can make this challenging. To that end, we derive an equivalent Lagrangian formulation of (6), where we "move" the constraints into the objective function, so that we can define a margin between the expected reward of two sets of policies that captures their "difference".

### 3.1 Equivalent constraints via temporal difference learning

Intuitively, the Nash equilibrium constraints imply that any agent $i$ cannot improve $\pi_i$ via 1-step temporal difference learning; if the condition for Equation (3) is not satisfied for some $v_i$, $q_i$, and $(s, a_i)$, this would suggest that we can update the policy for agent $i$ and its value function. Based on this notion, we can derive equivalent versions of the constraints corresponding to $t$-step temporal difference (TD) learning.

**Theorem 1.** *For a certain policy $\boldsymbol{\pi}$ and reward $\mathbf{r}$, let $\hat{v}_i(s; \boldsymbol{\pi}, \mathbf{r})$ be the unique solution to the Bellman equation:*

$$\hat{v}_i(s; \boldsymbol{\pi}, \mathbf{r}) = \mathbb{E}_{\mathbf{a} \sim \boldsymbol{\pi}} \left[ r_i(s, \mathbf{a}) + \gamma \sum_{s' \in \mathcal{S}} P(s'|s, \mathbf{a}) \hat{v}_i(s'; \boldsymbol{\pi}, \mathbf{r}) \right] \qquad \forall s \in \mathcal{S}.$$

*Denote $\hat{q}_i^{(t)}(\{s^{(j)}, \mathbf{a}^{(j)}\}_{j=0}^{t-1}, s^{(t)}, a_i^{(t)}; \boldsymbol{\pi}, \mathbf{r})$ as the discounted expected return for the $i$-th agent conditioned on visiting the trajectory $\{s^{(j)}, \mathbf{a}^{(j)}\}_{j=0}^{t-1}, s^{(t)}$ in the first $(t-1)$ steps and choosing action $a_i^{(t)}$ at the $t$ step, when other agents use policy $\pi_{-i}$:*

$$\hat{q}_i^{(t)}(\{s^{(j)}, \mathbf{a}^{(j)}\}_{j=0}^{t-1}, s^{(t)}, a_i^{(t)}; \boldsymbol{\pi}, \mathbf{r})$$
$$= \sum_{j=0}^{t-1} \gamma^j r_i(s^{(j)}, a^{(j)}) + \gamma^t \mathbb{E}_{a_{-i} \sim \pi_{-i}} \left[ r_i(s^{(t)}, \mathbf{a}^{(t)}) + \gamma \sum_{s' \in \mathcal{S}} P(s'|s, \mathbf{a}^{(t)}) \hat{v}_i(s'; \boldsymbol{\pi}, \mathbf{r}) \right].$$

*Then $\boldsymbol{\pi}$ is Nash equilibrium if and only if for all $t \in \mathbb{N}^+, i \in [N], j \in [t], s^{(j)} \in \mathcal{S}, a^{(j)} \in \mathcal{A}$*

$$\hat{v}_i(s^{(0)}; \boldsymbol{\pi}, \mathbf{r}) \geq \mathbb{E}_{a_{-i} \sim \pi_{-i}} \left[ \hat{q}_i^{(t)}(\{s^{(j)}, \mathbf{a}^{(j)}\}_{j=0}^{t-1}, s^{(t)}, a_i^{(t)}; \boldsymbol{\pi}, \mathbf{r}) \right] \triangleq Q_i^{(t)}(\{s^{(j)}, a_i^{(j)}\}_{j=0}^t; \boldsymbol{\pi}, \mathbf{r}).$$
$$\tag{7}$$

Intuitively, Theorem 1 states that if we replace the 1-step constraints with $(t+1)$-step constraints, we obtain the same solution as MARL($\mathbf{r}$), since $(t+1)$-step TD updates (over one agent at a time) are still stationary with respect to a Nash equilibrium solution. So the constraints can be unrolled for $t$ steps and rewritten as $\hat{v}_i(s^{(0)}) \geq Q_i^{(t)}(\{s^{(j)}, a_i^{(j)}\}_{j=0}^t; \boldsymbol{\pi}, \mathbf{r})$ (corresponding to Equation (7)).

## 3.2 Multi-agent inverse reinforcement learning

We are now ready to construct the Lagrangian dual of the primal in Equation (6), using the equivalent formulation from Theorem 1. The first observation is that for any policy $\boldsymbol{\pi}$, $f(\boldsymbol{\pi}, \hat{v}) = 0$ when $\hat{v}$ is defined as in Theorem 1 (see Lemma 1 in appendix). Therefore, we only need to consider the "unrolled" constraints from Theorem 1, obtaining the following dual problem

$$\max_{\lambda \geq 0} \min_{\boldsymbol{\pi}} L_{\mathbf{r}}^{(t+1)}(\boldsymbol{\pi}, \lambda) \triangleq \sum_{i=1}^{N} \sum_{\tau_i \in \mathcal{T}_i^t} \lambda(\tau_i) \left( Q_i^{(t)}(\tau_i; \boldsymbol{\pi}, \mathbf{r}) - \hat{v}_i(s^{(0)}; \boldsymbol{\pi}, \mathbf{r}) \right) \tag{8}$$

where $\mathcal{T}_i(t)$ is the set of all length-$t$ trajectories of the form $\{s^{(j)}, a_i^{(j)}\}_{j=0}^{t}$, with $s^{(0)}$ as initial state, $\lambda$ is a vector of $N \cdot |\mathcal{T}_i(t)|$ Lagrange multipliers, and $\hat{v}$ is defined as in Theorem 1. This dual formulation is a sum over agents and trajectories, which uniquely corresponds to the constraints in Equation 7.

In the following theorem, we show that for a specific choice of $\lambda$ we can recover the difference of the sum of expected rewards between two policies, a performance gap similar to the one used in single agent IRL in Equation (4). This amounts to "relaxing" the primal problem.

**Theorem 2.** *For any two policies $\boldsymbol{\pi}^\star$ and $\boldsymbol{\pi}$, let*

$$\lambda_{\boldsymbol{\pi}}^\star(\tau_i) = \eta(s^{(0)})\pi_i(a_i^{(0)}|s^{(0)}) \prod_{j=1}^{t} \pi_i(a_i^{(j)}|s^{(j)}) \sum_{a_{-i}^{(j-1)}} P(s^{(j)}|s^{(j-1)}, a^{(j-1)})\pi_{-i}^\star(a_{-i}^{(j)}|s^{(j)})$$

*be the probability of generating the sequence $\tau_i$ using policy $\pi_i$ and $\pi_{-i}^\star$. Then*

$$\lim_{t \to \infty} L_r^{(t+1)}(\boldsymbol{\pi}^\star, \lambda_{\boldsymbol{\pi}}^\star) = \sum_{i=1}^{N} \mathbb{E}_{\pi_i, \pi_{-i}^\star}[r_i(s, a)] - \sum_{i=1}^{N} \mathbb{E}_{\pi_i^\star, \pi_{-i}^\star}[r_i(s, a)] \tag{9}$$

*where $L_r^{(t+1)}(\boldsymbol{\pi}^\star, \lambda_{\boldsymbol{\pi}}^\star)$ corresponds to the dual function where the multipliers are the probability of generating their respective trajectories of length $t$.*

We provide a proof in Appendix A.3. Intuitively, the $\lambda^\star(\tau_i)$ weights correspond to the probability of generating trajectory $\tau_i$ when the policy is $\pi_i$ for agent $i$ and $\pi_{-i}^\star$ for the other agents. As $t \to \infty$, the first term of left hand side in Equation (8), $\sum_{i=1}^{N} \sum_{\tau_i \in \mathcal{T}_i^t} \lambda(\tau_i)Q_i^{(t)}(\tau_i)$, converges to the expected total reward $\mathbb{E}_{\pi_i, \pi_{-i}^\star}[r_i]$, which is the first term of right hand side. The marginal of $\lambda^\star$ over the initial state is the initial state distribution, so the second term of left hand side, $\sum_s \hat{v}(s)\eta(s)$, converges to $\mathbb{E}_{\pi_i^\star, \pi_{-i}^\star}[r_i]$, which is the second term of right hand side. Thus, the left hand side and right hand side of Equation (8) are the same as $t \to \infty$. We could also view the right hand side of Equation (8) as the case where policies of $\pi_{-i}^\star$ are part of the environment.

Theorem 2 motivates the following definition of multi-agent IRL with regularizer $\psi$.

$$\text{MAIRL}_\psi(\boldsymbol{\pi}_E) = \arg\max_{\mathbf{r}} -\psi(\mathbf{r}) + \sum_{i=1}^{N}(\mathbb{E}_{\boldsymbol{\pi}_E}[r_i]) - \left(\max_{\boldsymbol{\pi}} \sum_{i=1}^{N}(\beta H_i(\pi_i) + \mathbb{E}_{\pi_i, \pi_{E_{-i}}}[r_i])\right),$$

where $H_i(\pi_i) = \mathbb{E}_{\pi_i, \pi_{E_{-i}}}[-\log \pi_i(a_i|s)]$ is the discounted causal entropy for policy $\pi_i$ when other agents follow $\pi_{E_{-i}}$, and $\beta$ is a hyper-parameter controlling the strength of the entropy regularization term as in [16]. This formulation is a strict generalization to the single agent IRL in [16].

**Corollary 2.1.** *If $N = 1$, $\beta = 1$ then $\text{MAIRL}_\psi(\boldsymbol{\pi}_E) = \text{IRL}_\psi(\boldsymbol{\pi}_E)$.*

Furthermore, if the regularization $\psi$ is additively separable, and for each agent $i$, $\pi_{E_i}$ is the unique optimal response to other experts $\pi_{E_{-i}}$, we obtain the following:

**Theorem 3.** *Assume that $\psi(\mathbf{r}) = \sum_{i=1}^{N} \psi_i(r_i)$, $\psi_i$ is convex for each $i \in [N]$, and that $\text{MARL}(\mathbf{r})$ has a unique solution[2] for all $r \in \text{MAIRL}_\psi(\boldsymbol{\pi}_E)$, then*

$$\text{MARL} \circ \text{MAIRL}_\psi(\boldsymbol{\pi}_E) = \arg\min_{\pi \in \Pi} \sum_{i=1}^{N} -\beta H_i(\pi_i) + \psi_i^\star(\rho_{\pi_{i, E_{-i}}} - \rho_{\pi_E}) \tag{10}$$

*where $\pi_{i, E_{-i}}$ denotes $\pi_i$ for agent $i$ and $\pi_{E_{-i}}$ for other agents.*

The above theorem suggests that $\psi$-regularized multi-agent inverse reinforcement learning is seeking, for each agent $i$, a policy whose occupancy measure is close to one where we replace policy $\pi_i$ with expert $\pi_{E_i}$, as measured by the convex function $\psi_i^\star$.

However, we do not assume access to the expert policy $\pi_E$ during training, so it is not possible to obtain $\rho_{\pi_{i,E_{-i}}}$. Therefore, we consider an alternative approach where we match the occupancy measure between $\rho_{\pi_E}$ and $\rho_\pi$. We can obtain our practical algorithm if we select an adversarial reward function regularizer and remove the effect from entropy regularizers.

**Proposition 2.** *If $\beta = 0$, and $\psi(\mathbf{r}) = \sum_{i=1}^N \psi_i(r_i)$ where $\psi_i(r_i) = \mathbb{E}_{\boldsymbol{\pi}_E}[g(r_i)]$ if $r_i > 0$; $+\infty$ otherwise, and*

$$g(x) = \begin{cases} -x - \log(1 - e^x) & \text{if } \quad r_i > 0 \\ +\infty & \text{otherwise} \end{cases}$$

*then*

$$\arg\min_\pi \sum_{i=1}^N \psi_i^\star(\rho_{\pi_i,\pi_{E_{-i}}} - \rho_{\boldsymbol{\pi}_E}) = \arg\min_\pi \sum_{i=1}^N \psi_i^\star(\rho_{\pi_i,\pi_{-i}} - \rho_{\boldsymbol{\pi}_E}) \tag{11}$$

*and both are equal to $\pi_E$.*

Theorem 3 and Proposition 2 discuss the differences from the single agent scenario. In Theorem 3 we make the assumption that MARL($\mathbf{r}$) has a unique solution, which is always true in the single agent case due to convexity of the space of the optimal policies. In Proposition 2 we remove the entropy regularizer because here the causal entropy for $\pi_i$ may depend on the policies of the other agents. Specifically, the entropy for the left hand side of Equation (11) conditions on $\pi_{E_{-i}}$ and the entropy for the right hand side conditions on $\pi_{-i}$ (both would disappear in the single-agent case).

## 4 Practical multi-agent imitation learning

Despite the recent successes in deep RL, it is notoriously hard to train policies with RL algorithms-because of high variance gradient estimates. This is further exacerbated in Markov games since an agent's optimal policy depends on other agents [14, 17]. In this section, we address these problems and propose practical algorithms for multi-agent imitation.

### 4.1 Multi-agent generative adversarial imitation learning

We select $\psi_i$ to be our reward function regularizer in Proposition 2; this corresponds to the two-player game introduced in Generative Adversarial Imitation Learning (GAIL, [16]). For each agent $i$, we have a discriminator (denoted as $D_{\omega_i}$) mapping state action-pairs to *scores* optimized to discriminate expert demonstrations from behaviors produced by $\pi_i$. Implicitly, $D_{\omega_i}$ plays the role of a reward function for the generator, which in turn attempts to train the agent to maximize its reward thus fooling the discriminator. We optimize the following objective:

$$\min_\theta \max_\omega \mathbb{E}_{\boldsymbol{\pi}_\theta}\left[\sum_{i=1}^N \log D_{\omega_i}(s, a_i)\right] + \mathbb{E}_{\boldsymbol{\pi}_E}\left[\sum_{i=1}^N \log(1 - D_{\omega_i}(s, a_i))\right] \tag{12}$$

We update $\pi_\theta$ through reinforcement learning, where we also use a baseline $V_\phi$ to reduce variance. We outline the algorithm – Multi-Agent GAIL (MAGAIL) – in Appendix B.

We can augment the reward regularizer $\psi(\mathbf{r})$ using an indicator $y(r)$ denoting whether $r$ fits our prior knowledge; the augmented reward regularizer $\hat{\psi} : \mathbb{R}^{\mathcal{S} \times \mathcal{A}} \to \mathbb{R} \cup \{\infty\}$ is then: $\psi(\mathbf{r})$ if $y(\mathbf{r}) = 1$ and $\infty$ if $y(\mathbf{r}) = 0$. We introduce three types of $y(\mathbf{r})$ for common settings.

**Centralized** The easiest case is to assume that the agents are fully cooperative, i.e. they share the same reward function. Here $y(\mathbf{r}) = \mathbb{I}(r_1 = r_2 = \ldots r_n)$ and $\psi(\mathbf{r}) = \psi_{\mathrm{GA}}(\mathbf{r})$. One could argue this corresponds to the GAIL case, where the RL procedure operates on multiple agents (a joint policy).

**Decentralized** We make no prior assumptions over the correlation between the rewards. Here $y(\mathbf{r}) = \mathbb{I}(r_i \in \mathbb{R}^{\mathcal{O}_i \times \mathcal{A}_i})$ and $\psi_i(r_i) = \psi_{\mathrm{GA}}(r_i)$. This corresponds to one discriminator for each agent which discriminates the trajectories as observed by agent $i$. However, these discriminators are not learned independently as they interact indirectly via the environment.

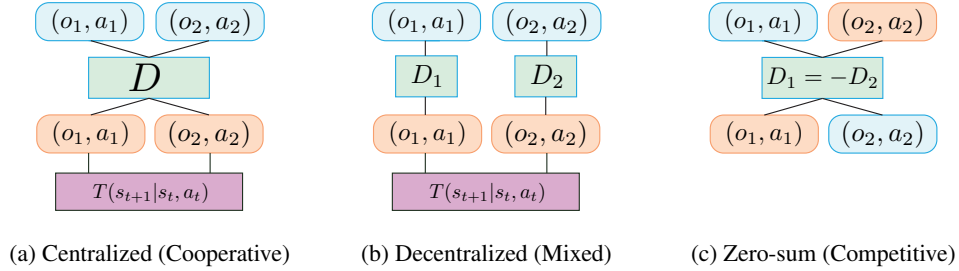

|  (a) Centralized (Cooperative) | (b) Decentralized (Mixed) | (c) Zero-sum (Competitive) |

Figure 1: Different MAGAIL algorithms obtained with different priors on the reward structure. The discriminator tries to assign higher rewards to top row and low rewards to bottom row. In centralized and decentralized, the policy operates with the environment to match the expert rewards. In zero-sum, the policy do not interact with the environment; expert and policy trajectories are paired together as input to the discriminator.

**Zero Sum**  Assume there are two agents that receive opposite rewards, so $r_1 = -r_2$. As such, $\psi$ is no longer additively separable. Nevertheless, an adversarial training procedure can be designed using the following fact:

$$v(\pi_{E_1}, \pi_2) \geq v(\pi_{E_1}, \pi_{E_2}) \geq v(\pi_1, \pi_{E_2}) \tag{13}$$

where $v(\pi_1, \pi_2) = \mathbb{E}_{\pi_1, \pi_2}[r_1(s, a)]$ is the expected outcome for agent 1, and is modeled by the discriminator. The discriminator could then try to maximize $v$ for trajectories from $(\pi_{E_1}, \pi_2)$ and minimize $v$ for trajectories from $(\pi_2, \pi_{E_1})$ according to Equation (13).

These three settings are in summarized in Figure 1.

### 4.2  Multi-agent actor-critic with Kronecker factors

To optimize over the generator parameters $\theta$ in Eq. (12) we wish to use an algorithm for multi-agent RL that has good sample efficiency in practice. Our algorithm, which we refer to as Multi-agent Actor-Critic with Kronecker-factors (MACK), is based on Actor-Critic with Kronecker-factored Trust Region (ACKTR, [25–27]), a state-of-the-art natural policy gradient [28, 29] method in deep RL. MACK uses the framework of centralized training with decentralized execution [17]; policies are trained with additional information to reduce variance but such information is not used during execution time. We let the advantage function of every agent agent be a function of all agents' observations and actions:

$$A^{\pi_i}_{\phi_i}(s, \mathbf{a}_t) = \sum_{j=0}^{k-1} (\gamma^j r_i(s_{t+j}, \mathbf{a}_{t+j}) + \gamma^k V^{\pi_i}_{\phi_i}(s_{t+k}, a_{-i,t+k})) - V^{\pi_i}_{\phi_i}(s_t, a_{-i,t}) \tag{14}$$

where $V^{\pi_i}_{\phi_i}(s_k, a_{-i})$ is the baseline for $i$, utilizing the additional information $(a_{-i})$ for variance reduction. We use (approximated) natural policy gradients to update both $\theta$ and $\phi$ but without trust regions to schedule the learning rate, using a linear decay learning rate schedule instead.

MACK has some notable differences from Multi-Agent Deep Deterministic Policy Gradient [14]. On the one hand, MACK does not assume knowledge of other agent's policies nor tries to infer them; the value estimator merely collects experience from other agents (and treats them as black boxes). On the other hand, MACK does not require gradient estimators such as Gumbel-softmax [30, 31] to optimize over discrete actions, which is necessary for DDPG [32].

## 5  Experiments

We evaluate the performance of (centralized, decentralized, and zero-sum versions) of MAGAIL under two types of environments. One is a particle environment which allows for complex interactions and behaviors; the other is a control task, where multiple agents try to cooperate and move a plank forward. We collect results by averaging over 5 random seeds. Our implementation is based on OpenAI baselines [33]; please refer to Appendix C for implementation details[3].

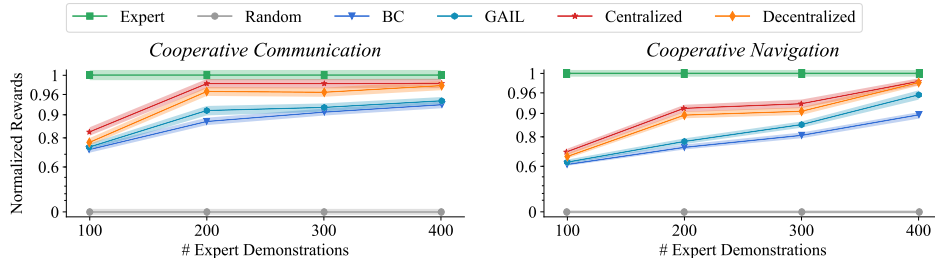

Figure 2: Average true reward from cooperative tasks. Performance of experts and random policies are normalized to one and zero respectively. We use inverse log scale for better comparison.

We compare our methods (centralized, decentralized, zero-sum MAGAIL) with two baselines. The first is behavior cloning (BC), which learns a maximum likelihood estimate for $a_i$ given each state $s$ and does not require actions from other agents. The second baseline is a GAIL IRL baseline that operates on each agent separately – for each agent we first pretrain the other agents with BC, and then train the agent with GAIL; we then gather the trained GAIL policies from all the agents and evaluate their performance.

## 5.1 Particle environments

We first consider the particle environment proposed in [14], which consists of several agents and landmarks. We consider two cooperative environments and two competitive ones. All environments have an underlying true reward function that allows us to evaluate the performance of learned agents.

The environments include: **Cooperative Communication** – two agents must cooperate to reach one of three colored landmarks. One agent ("speaker") knows the goal but cannot move, so it must convey the message to the other agent ("listener") that moves but does not observe the goal. **Cooperative Navigation** – three agents must cooperate through physical actions to reach three landmarks; ideally, each agent should cover a single landmark. **Keep-Away** – two agents have contradictory goals, where agent 1 tries to reach one of the two targeted landmarks, while agent 2 (the adversary) tries to keep agent 1 from reaching its target. The adversary does not observe the target, so it must act based on agent 1's actions. **Predator-Prey** – three slower cooperating adversaries must chase the faster agent in a randomly generated environment with obstacles; the adversaries are rewarded by touching the agent while the agent is penalized.

For the cooperative tasks, we use an analytic expression defining the expert policy; for the competitive tasks, we use MACK to train expert policies based on the true underlying rewards (using larger policy and value networks than the ones that we use for imitation). We then use the expert policies to simulate trajectories $\mathcal{D}$, and then do imitation learning on $\mathcal{D}$ as demonstrations, where we assume the underlying rewards are unknown. Following [34], we pretrain our Multi-Agent GAIL methods and the GAIL baseline using behavior cloning as initialization to reduce sample complexity for exploration. We consider 100 to 400 episodes of expert demonstrations, each with 50 timesteps, which is close to the amount of timesteps used for the control tasks in [16]. Moreover, we randomly sample the starting position of agent and landmarks each episode, so our policies have to learn to generalize when they encounter new settings.

### 5.1.1 Cooperative tasks

We evaluate performance in cooperative tasks via the average expected reward obtained by all the agents in an episode. In this environment, the starting state is randomly initialized, so generalization is crucial. We do not consider the zero-sum case, since it violates the cooperative nature of the task. We display the performance of centralized, decentralized, GAIL and BC in Figure 2.

Naturally, the performance of BC and MAGAIL increases with more expert demonstrations. MA-GAIL performs consistently better than BC in all the settings; interestingly, in the cooperative communication task, centralized MAGAIL is able to achieve expert-level performance with only 200 demonstrations, but BC fails to come close even with 400 trajectories. Moreover, the centralized MA-

Table 1: Average agent rewards in competitive tasks. We compare behavior cloning (BC), GAIL (G), Centralized (C), Decentralized (D), and Zero-Sum (ZS) methods. Best marked in bold (high vs. low rewards is preferable depending on the agent vs. adversary role).

| Task | Predator-Prey | | | | | | | | |
|---|---|---|---|---|---|---|---|---|---|
| Agent | Behavior Cloning | | | | | G | C | D | ZS |
| Adversary | BC | G | C | D | ZS | Behavior Cloning | | | |
| Rewards | -93.20 | -93.71 | -93.75 | -95.22 | **-95.48** | -90.55 | -91.36 | **-85.00** | -89.4 |
| Task | Keep-Away | | | | | | | | |
| Agent | Behavior Cloning | | | | | G | C | D | ZS |
| Adversary | BC | G | C | D | ZS | Behavior Cloning | | | |
| Rewards | 24.22 | 24.04 | 23.28 | 23.56 | **23.19** | 26.22 | 26.61 | **28.73** | 27.80 |

GAIL performs slightly better than decentralized MAGAIL due to the better prior, but decentralized MAGAIL still learns a highly correlated reward between two agents.

### 5.1.2 Competitive tasks

We consider all three types of Multi-Agent GAIL (centralized, decentralized, zero-sum) and BC in both competitive tasks. Since there are two opposing sides, it is hard to measure performance directly. Therefore, we compare by letting (agents trained by) BC play against (adversaries trained by) other methods, and vice versa. From Table 1, decentralized and zero-sum MAGAIL often perform better than centralized MAGAIL and BC, which suggests that the selection of the suitable prior $\hat{\psi}$ is important for good empirical performance.

### 5.2 Cooperative control

In some cases we are presented with sub-optimal expert demonstrations because the environment has changed; we consider this case in a cooperative control task [35], where $N$ bipedal walkers cooperate to move a long plank forward; the agents have incentive to collaborate since the plank is much longer than any of the agents. The expert demonstrates its policy on an environment with no bumps on the ground and heavy weights, while we perform imitation in an new environment with bumps and lighter weights (so one is likely to use too much force). Agents trained with BC tend to act more aggressively and fail, whereas agents trained with centralized MAGAIL can adapt to the new environment. With 10 (imperfect) expert demonstrations, BC agents have a chance of failure of $39.8\%$ (with a reward of 1.26), while centralized MAGAIL agents fail only $26.2\%$ of the time (with a reward of 26.57). We show videos of respective policies in the supplementary.

## 6 Discussion

There is a vast literature on single-agent imitation learning [36]. Behavior Cloning (BC) learns the policy through supervised learning [37]. Inverse Reinforcement Learning (IRL) assumes the expert policy optimizes over some unknown reward, recovers the reward, and learns the policy through reinforcement learning (RL). BC does not require knowledge of transition probabilities or access to the environment, but suffers from compounding errors and covariate shift [38, 23].

Most existing work in multi-agent imitation learning assumes the agents have very specific reward structures. The most common case is fully cooperative agents [39], where the challenges mainly lie in other factors, such as unknown role assignments [40], scalability to swarm systems [41] and agents with partial observations [42]. In non-cooperative settings, [43] consider the case of IRL for two-player zero-sum games and cast the IRL problem as Bayesian inference, while [44] assume agents are non-cooperative but the reward function is a linear combination of pre-specified features.

Our work is the first to propose a general multi-agent IRL framework that combines state-of-the art multi-agent reinforcement learning methods [14, 17] and implicit generative models such as generative adversarial networks [45]. Experimental results demonstrate that it is able to imitate complex behaviors in high-dimensional environments with both cooperative and adversarial interactions. An interesting future direction is to explore new paradigms for learning from experts, such as allowing the expert to participate in the agent's learning process [24].

**Acknowledgements**

This work was supported by Toyota Research Institute and Future of Life Institute. The authors would like to thank Lantao Yu for discussions over implementation.

## Footnotes

[1]We use the term "entropy" to denote the $\gamma$-discounted causal entropy for policies in the rest of the paper.

[2]The set of Nash equilibria is not always convex, so we have to assume $\text{MARL}(\mathbf{r})$ returns a unique solution.

[3]Code for reproducing the experiments are in `https://github.com/ermongroup/multiagent-gail`.

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
