[Supplementary Material · magail_supp.pdf]

# A  Proofs

We use $\hat{v}_i(s)$, $\hat{q}_i(s, a_i)$ and $Q(\tau)$ to represent $\hat{v}_i(s; \boldsymbol{\pi}, \mathbf{r})$, $\hat{q}_i(s, a_i; \boldsymbol{\pi}, \mathbf{r})$ and $Q(\tau; \boldsymbol{\pi}, \mathbf{r})$, where we implicitly assume dependency over $\boldsymbol{\pi}$ and $\mathbf{r}$.

## A.1  Proof to Lemma 1

For any policy $\boldsymbol{\pi}$, $f_{\mathbf{r}}(\boldsymbol{\pi}, \hat{v}) = 0$ when $\hat{v}$ is the value function of $\boldsymbol{\pi}$ (due to Bellman equations). However, only policies that form a Nash equilibrium satisfies the constraints in Eq. 2; we formalize this in the following Lemma.

**Lemma 1.** *Let $\hat{v}_i(s; \boldsymbol{\pi}, \mathbf{r})$ be the solution to the Bellman equation*

$$\hat{v}_i(s) = \mathbb{E}_{\mathbf{a} \sim \boldsymbol{\pi}}[r_i(s, \mathbf{a}) + \gamma \sum_{s' \in \mathcal{S}} P(s'|s, \mathbf{a})\hat{v}_i(s')]$$

*and $\hat{q}_i(s, a_i) = \mathbb{E}_{a_{-i} \sim \pi_{-i}}[r_i(s, \mathbf{a}) + \gamma \sum_{s' \in \mathcal{S}} P(s'|s, \mathbf{a})\hat{v}_i(s')]$. Then for any $\pi$,*

$$f_{\mathbf{r}}(\boldsymbol{\pi}, \hat{v}(\boldsymbol{\pi})) = 0$$

*Furthermore, $\boldsymbol{\pi}$ is Nash equilibrium under $r$ if and only if $\hat{v}_i(s) \geq \hat{q}_i(s, a_i)$ for all $i \in [N], s \in \mathcal{S}, a_i \in \mathcal{A}_i$.*

*Proof.* By definition of $\hat{v}_i(s)$ we have:

$$\hat{v}_i(s) = \mathbb{E}_{\mathbf{a} \sim \boldsymbol{\pi}}[r_i(s, \mathbf{a}) + \gamma \sum_{s' \in \mathcal{S}} P(s'|s, \mathbf{a})\hat{v}_i(s')]$$

$$= \mathbb{E}_{a_i \sim \pi_i} \mathbb{E}_{a_{-i} \sim \pi_{-i}}[r_i(s, \mathbf{a}) + \gamma \sum_{s' \in \mathcal{S}} P(s'|s, \mathbf{a})\hat{v}_i(s')]$$

$$= \mathbb{E}_{a_{-i} \sim \pi_i}[\hat{q}_i(s, a_i)]$$

which uses the fact that $a_i$ and $a_{-i}$ are independent conditioned on $s$. Hence $f_{\mathbf{r}}(\boldsymbol{\pi}, \hat{v}) = 0$ immediately follows.

If $\boldsymbol{\pi}$ is a Nash equilibrium, and at least one of the constrains does not hold, i.e. there exists some $i$ and $s, a_i$ such that $\hat{v}(s) < \hat{q}(s, a_i)$, then agent $i$ can achieve a strictly higher expected return if it chooses to take actions $a_i$ whenever it encounters state $s_i$ and follow $\pi_i$ for rest of the states, which violates the Nash equilibrium assumption.

If the constraints hold, i.e. for all $i$ and $(s, a_i)$, $\hat{v}_i(s) \geq \hat{q}_i(s, a_i)$ then

$$\hat{v}_i(s) \geq \mathbb{E}_{\pi_i}[\hat{q}_i(s, a_i)] = \hat{v}_i(s)$$

so value iteration over $\hat{v}_i(s)$ converges. If we can find another policy $\boldsymbol{\pi}'$ such that $\hat{v}_i(s) < \mathbb{E}_{\pi'_i}[\hat{q}_i(s, a_i)]$, then there should be at least one violation in the constraints since $\pi'_i$ must be a convex combination (expectation) over actions $a_i$. Therefore, for any policy $\pi'_i$ and action $a_i$ for any agent $i$, $\mathbb{E}_{\pi_i}[\hat{q}_i(s, a_i)] \geq \mathbb{E}_{\pi'_i}[\hat{q}_i(s, a_i)]$ always hold, so $\pi_i$ is the optimal response to $\pi_{-i}$, and $\boldsymbol{\pi}$ constitutes a Nash equilibrium when we repeat this argument for all agents.

Notably, Theorem 3.8.2 in [21] discusses the equivalence by assuming $f_{\mathbf{r}}(\boldsymbol{\pi}, v) = 0$ for some $v$; if $v$ satisfies the assumptions, then $v = \hat{v}'$. $\qquad\square$

## A.2  Proof to Theorem 1

*Proof.* If $\pi$ is a Nash equilibrium, and at least one of the constraints does not hold, i.e. there exists some $i$ and $\{s^{(j)}, a_i^{(j)}\}_{j=0}^{t}$, such that

$$\hat{v}_i(s^{(0)}) < \mathbb{E}_{\pi_{-i}}[\hat{q}_i^{(t)}(\{s^{(j)}, \mathbf{a}^{(j)}\}_{j=0}^{t-1}, s^{(t)}, a_i^{(t)})]$$

Then agent $i$ can achieve a strictly higher expected return on its own if it chooses a particular sequence of actions by taking $a_i^{(j)}$ whenever it encounters state $s^{(j)}$, and follow $\pi_i$ for the remaining states. We note that this is in expectation over the policy of other agents. Hence, we construct a policy for agent

$i$ that has strictly higher value than $\pi_i$ without modifying $\pi_{-i}$, which contradicts the definition of Nash equilibrium.

If the constraints hold, i.e for all $i$ and $\{s^{(j)}, a_i^{(j)}\}_{j=0}^t$,

$$\hat{v}_i(s^{(0)}) \geq \mathbb{E}_{\pi_{-i}}[\hat{q}_i^{(t)}(\{s^{(j)}, \mathbf{a}^{(j)}\}_{j=0}^{t-1}, s^{(t)}, a_i^{(t)})]$$

then we can construct any $\hat{q}_i(s^{(0)}, a_i^{(0)})$ via a convex combination by taking the expectation over $\pi_i$:

$$\hat{q}_i(s^{(0)}, a_i^{(0)}) = \mathbb{E}_{\pi_i}[\mathbb{E}_{\pi_{-i}}[\hat{q}_i^{(t)}(\{s^{(j)}, \mathbf{a}^{(j)}\}_{j=0}^{t-1}, s^{(t)}, a_i^{(t)})]]$$

where the expectation over $\pi_i$ is taken over actions $\{a_i^{(j)}\}_{j=0}^t$ (the expectation over states are contained in the inner expectation over $\pi_{-i}$). Therefore, $\forall i \in [N], s \in \mathcal{S}, a_i \in \mathcal{A}_i$,

$$\hat{v}_i(s) \geq \hat{q}_i(s, a_i)$$

and we recover the constraints in Eq. 2. By Lemma 1, $\pi$ is a Nash equilibrium. $\qquad\square$

### A.3 Proof to Theorem 2

*Proof.* We use $Q^\star, \hat{q}^\star, \hat{v}^\star$ to denote the $Q, \hat{q}$ and $\hat{v}$ quantities defined for policy $\boldsymbol{\pi}^\star$. For the two terms in $L_{\mathbf{r}}^{(t+1)}(\boldsymbol{\pi}^\star, \lambda_{\boldsymbol{\pi}}^\star)$ we have:

$$L_{\mathbf{r}}^{(t+1)}(\boldsymbol{\pi}^\star, \lambda_{\boldsymbol{\pi}}^\star) = \sum_{i=1}^N \sum_{\tau_i \in \mathcal{T}_i} \lambda^\star(\tau_i)(Q_i^\star(\tau_i) - \hat{v}_i^\star(s^{(0)})) \qquad (15)$$

For any agent $i$, we note that

$$\sum_{\tau_i \in \mathcal{T}_i} \lambda^\star(\tau_i)Q_i^\star(\tau_i) = \mathbb{E}_{\pi_i}\mathbb{E}_{\pi_{-i}^\star}[\sum_{j=0}^{t-1} \gamma^j r_i(s^{(j)}, a^{(j)}) + \gamma^t \hat{q}_i^\star(s^t, a_i^{(t)})]$$

which amounts to using $\pi_i$ for agent $i$ for the first $t$ steps and using $\pi_i^\star$ for the remaining steps, whereas other agents follow $\pi_{-i}^\star$. As $t \to \infty$, this converges to $\mathbb{E}_{\pi_i, \pi_{-i}^\star}[r_i]$ since $\gamma^t \to 0$ and $q_i^\star(s^{(t)}, a_i^{(t)})$ is bounded. Moreover, for $\hat{v}_i^\star(s^{(0)})$, we have

$$\sum_{\tau_i \in \mathcal{T}_i} \lambda^\star(\tau_i)\hat{v}_i^\star(s^{(0)}) = \mathbb{E}_{s^{(0)} \sim \eta}[\hat{v}_i^\star(s^{(0)})] = \mathbb{E}_{\boldsymbol{\pi}^\star}[r_i]$$

Combining the two we have

$$L_{\mathbf{r}}^{(t+1)}(\boldsymbol{\pi}^\star, \lambda_{\boldsymbol{\pi}}^\star) = \sum_{i=1}^N \mathbb{E}_{\pi_i, \pi_{-i}^\star}[r_i] - \sum_{i=1}^N \mathbb{E}_{\boldsymbol{\pi}^\star}[r_i]$$

which describes the differences in expected rewards. $\qquad\square$

### A.4 Proof to Theorem 3

*Proof.* Define the "MARL" objective for a single agent $i$ where other agents have policy $\pi_{E_i}$:

$$\text{MARL}_i(r_i) = \max_{\pi_i} H_i(\pi_i) + \mathbb{E}_{\pi_i, \pi_{E_{-i}}}[r_i]$$

Define the "MAIRL" objective for a single agent $i$ where other agents have policy $\boldsymbol{\pi}_E$:

$$\text{MAIRL}_{i,\psi}(\boldsymbol{\pi}^\star) = \arg\max_{r_i} \psi_i(r_i) + \mathbb{E}_{\boldsymbol{\pi}_E}[r_i] - (\max_{\pi_i} H_i(\pi_i) + \mathbb{E}_{\pi_i, \pi_{E_{-i}}}[r_i])$$

Since $r_i$ and $\pi_i$'s are independent in the MAIRL objective, the solution to $\text{MAIRL}_\psi$ can be represented by the solutions of $\text{MAIRL}_{i,\psi}$ for each $i$:

$$\text{MAIRL}_\psi = [\text{MAIRL}_{1,\psi}, \dots, \text{MAIRL}_{N,\psi}]$$

Moreover, the single agent "MARL" objective $\text{MARL}_i(r_i)$ has a unique solution $\pi_{E_i}$, which also composes the (unique) solution to MARL (which we assumed in Section 3. Therefore,

$$\text{MARL}(\mathbf{r}) = [\text{MARL}_1(r_1), \dots, \text{MARL}_N(r_N)]$$

So we can use Proposition 3.1 in [16] for each agent $i$ with $\text{MARL}_i(r_i)$ and $\text{MAIRL}_{i,\psi}(\boldsymbol{\pi}^\star)$ and achieve the same solution as $\text{MARL} \circ \text{MAIRL}_\psi$. $\qquad\square$

### A.5 Proof to Proposition 2

*Proof.* From Corollary A.1.1 in [16], we have

$$\psi_{GA}^{\star}(\rho_{\boldsymbol{\pi}} - \rho_{\boldsymbol{\pi}_E}) = \max_{D \in (0,1)^{S \times \mathcal{A}}} \mathbb{E}_{\boldsymbol{\pi}}[\log D(s,a)] + \mathbb{E}_{\boldsymbol{\pi}_E}[\log(1 - D(s,a))] \equiv D_{JS}(\rho_{\boldsymbol{\pi}}, \rho_{\boldsymbol{\pi}_E})$$

where $D_{JS}$ denotes Jensen-Shannon divergence (which is a squared metric), and $\equiv$ denotes equivalence up to shift and scaling.

Taking the min over this we obtain

$$\arg\min_{\pi} \sum_{i=1}^{N} \psi_{GA}^{\star}(\rho_{\boldsymbol{\pi}} - \rho_{\boldsymbol{\pi}_E}) = \boldsymbol{\pi}_E$$

Similarly,

$$\arg\min_{\pi} \sum_{i=1}^{N} \psi_{GA}^{\star}(\rho_{\pi_i, \pi_{E_{-i}}} - \rho_{\boldsymbol{\pi}_E}) = \boldsymbol{\pi}_E$$

So these two quantities are equal. □

## B MAGAIL Algorithm

We include the MAGAIL algorithm as follows:

---
**Algorithm 1** Multi-Agent GAIL (MAGAIL)

---
**Input:** Initial parameters of policies, discriminators and value (baseline) estimators $\theta_0, \omega_0, \phi_0$; expert trajectories $\mathcal{D} = \{(s_j, a_j)\}_{j=0}^{M}$; batch size $B$; Markov game as a black box $(N, \mathcal{S}, \mathcal{A}, \eta, T, r, \mathbf{o}, \gamma)$; initial policy $\boldsymbol{\pi}$.
**Output:** Learned policies $\pi_\theta$ and reward functions $D_\omega$.

---
**for** $u = 0, 1, 2, \ldots$ **do**
  Obtain trajectories of size $B$ from $\boldsymbol{\pi}$ by the process
  $$s_0 \sim \eta(s), \mathbf{a}_t \sim \pi_{\theta_u}(\mathbf{a}_t|s_t), s_{t+1} \sim P(s_t|\mathbf{a}_t)$$
  Sample state-action pairs from $\mathcal{D}$ with batch size $B$.
  Denote state-action pairs from $\pi$ and $\mathcal{D}$ as $\chi$ and $\chi_E$.
  **for** $i = 1, \ldots, n$ **do**
    Update $\omega_i$ to increase the objective
    $$\mathbb{E}_\chi[\log D_{\omega_i}(s, a_i)] + \mathbb{E}_{\chi_E}[\log(1 - D_{\omega_i}(s, a_i))]$$

  **end for**
  **for** $i = 1, \ldots, n$ **do**
    Compute value estimate $V^\star$ and advantage estimate $A_i$ for $(s, \mathbf{a}) \in \chi$.
    Update $\phi_i$ to decrease the objective
    $$\mathbb{E}_\chi[(V_\phi(s, a_{-i}) - V^\star(s, a_{-i}))^2]$$

    Update $\theta_i$ by policy gradient with small step sizes:
    $$\mathbb{E}_\chi[\nabla_{\theta_i} \pi_{\theta_i}(a_i|o_i) A_i(s, \mathbf{a})]$$

  **end for**
**end for**

---

Table 2: Performance in cooperative navigation.

| # Expert Episodes | 100 | 200 | 300 | 400 |
|---|---|---|---|---|
| Expert | | $-13.50 \pm 6.3$ | | |
| Random | | $-128.13 \pm 32.1$ | | |
| Behavior Cloning | $-56.82 \pm 18.9$ | $-43.10 \pm 16.0$ | $-35.66 \pm 15.2$ | $-25.83 \pm 12.7$ |
| Centralized | $\mathbf{-46.66} \pm 20.8$ | $\mathbf{-23.10} \pm 12.4$ | $\mathbf{-21.53} \pm 12.9$ | $\mathbf{-15.30} \pm 7.0$ |
| Decentralized | $-50.00 \pm 18.6$ | $-25.61 \pm 12.3$ | $-24.10 \pm 13.3$ | $-15.55 \pm 6.5$ |
| GAIL | $-55.01 \pm 17.7$ | $-39.21 \pm 16.5$ | $-29.89 \pm 13.5$ | $-18.76 \pm 12.1$ |

Table 3: Performance in cooperative communication.

| # Expert Episodes | 100 | 200 | 300 | 400 |
|---|---|---|---|---|
| Expert | | $-6.22 \pm 4.5$ | | |
| Random | | $-62.49 \pm 28.7$ | | |
| Behavior Cloning | $-21.25 \pm 10.6$ | $-13.25 \pm 7.4$ | $-11.37 \pm 5.9$ | $-10.00 \pm 5.36$ |
| Centralized | $\mathbf{-15.65} \pm 10.0$ | $\mathbf{-7.11} \pm 4.8$ | $\mathbf{-7.11} \pm 4.8$ | $\mathbf{-7.09} \pm 4.8$ |
| Decentralized | $-18.68 \pm 10.4$ | $-8.06 \pm 5.3$ | $-8.16 \pm 5.5$ | $-7.34 \pm 4.9$ |
| GAIL | $-20.28 \pm 10.1$ | $-11.06 \pm 7.8$ | $-10.51 \pm 6.6$ | $-9.44 \pm 5.7$ |

## C  Experiment Details

### C.1  Hyperparameters

For the particle environment, we use two layer MLPs with 128 cells in each layer, for the policy generator network, value network and the discriminator. We use a batch size of 1000. The policy is trained using K-FAC optimizer [46] with learning rate of 0.1. All other parameters for K-FAC optimizer are the same in [25].

For the cooperative control task, we use two layer MLPs with 64 cells in each layer for all the networks. We use a batch size of 2048, and learning rate of 0.03. We obtain expert trajectories by training the expert with MACK and sampling demonstrations from the same environment. Hence, the expert's demonstrations are imperfect (or even flawed) in the environment that we test on.

The particle environments are setup exactly as in OpenAI MultiAgent Particle Environment, except for two minor differences. One, we set the environment to have maximum episode length of 50. Two, we end the episode once the agents have reached their targets in the cooperative environments.

### C.2  Detailed Results

We use the particle environment introduced in [14] and the multi-agent control environment [35] for experiments. We list the exact performance in Tables 2, 3 for cooperative tasks, and Table 4 and competitive tasks. The means and standard deviations are computed over 100 episodes. The policies in the cooperative tasks are trained with varying number of expert demonstrations. The policies in the competitive tasks are trained with on a dataset with 100 expert trajectories.

The environment for each episode is drastically different (e.g. location of landmarks are randomly sampled), which leads to the seemingly high standrad deviation across episodes.

### C.3  Video Demonstrations

We show certain trajectories generated by our methods. The vidoes are here: videos[4].

For the particle case:

**Navigation-BC-Agents.gif**  Agents trained by behavior cloning in the navigation task.

**Navigation-GAIL-Agents.gif**  Agents trained by proposed framework in the navigation task.

**Predator-Prey-BC-Agent-BC-Adversary.gif**  Agent (green) trained by behavior cloning play against adversaries (red) trained by behavior cloning.

Table 4: Performance in competitive tasks.

| Task | Agent Policy | Adversary Policy | Agent Reward |
|---|---|---|---|
| Predator-Prey | Behavior Cloning | Behavior Cloning | $-93.20 \pm 63.7$ |
| | | GAIL | $-93.71 \pm 64.2$ |
| | | Centralized | $-93.75 \pm 61.9$ |
| | | Decentralized | $-95.22 \pm 49.7$ |
| | | Zero-Sum | $\mathbf{-95.48} \pm 50.4$ |
| | GAIL | Behavior Cloning | $-90.55 \pm 63.7$ |
| | Centralized | | $-91.36 \pm 58.7$ |
| | Decentralized | | $\mathbf{-85.00} \pm 42.3$ |
| | Zero-Sum | | $-89.4 \pm 48.2$ |
| Keep-Away | Behavior Cloning | Behavior Cloning | $24.22 \pm 21.1$ |
| | | GAIL | $24.04 \pm 18.2$ |
| | | Centralized | $23.28 \pm 20.6$ |
| | | Decentralized | $23.56 \pm 19.9$ |
| | | Zero-Sum | $\mathbf{23.19} \pm 19.9$ |
| | GAIL | Behavior Cloning | $26.22 \pm 19.1$ |
| | Centralized | | $26.61 \pm 20.0$ |
| | Decentralized | | $\mathbf{28.73} \pm 18.3$ |
| | Zero-Sum | | $27.80 \pm 19.2$ |

Figure 3: Sample complexity of multi-agent GAIL methods under cooperative tasks. Performance of experts is normalized to one, and performance of behavior cloning is normalized to zero. The standard deviation is computed with respect to episodes, and is noisy due to randomness in the environment.

**Predator-Prey-GAIL-Agent-BC-Adversary.gif** Agent (green) trained by proposed framework play against adversaries (red) trained by behavior cloning.

For the cooperative control case:

**Multi-Walker-Expert.mp4** Expert demonstrations in the "easy" environment.

**Multi-Walker-GAIL.mp4** Centralized GAIL trained on the "hard" environment.

**Multi-Walker-BC.mp4** BC trained on the "hard" environment.

Interestingly, the failure modes for the agents in "hard" environment is mostly having the plank fall off or bounce off, since by decreasing the weight of the plank will decrease its friction force and increase its acceleration.

### C.4 Potential Alternatives for RL Algorithms

There have been a range of deep reinforcement learning algorithms proposed recently; in this paper that all our MAGAIL / GAIL [16] algorithms in our experiments are using MACK / ACKTR [25]

as the underlying RL algorithm, so the performance gain for MAGAIL over GAIL is caused by the multi-agent formulation, instead of the specific RL algorithm used.

In fact, our MAGAIL formulation (similar to GAIL) does not restrict the choice of RL algorithm. We choose to use ACKTR as the RL algorithm because its scalability (as opposed to TRPO [47], which requires some form of inverse Fisher) and its ability to deal with discrete action spaces directly (as opposed to DDPG [14], which requires continuous relaxation of the discrete actions).

## Footnotes

[4]https://drive.google.com/open?id=1Oz4ezMaKiIsPUKtCEOb6YoHJ9jLk6zbj