[Reviews · NeurIPS 2018]

Reviewer 1



This paper proposed a novel framework for multiagent imitation learning for general Markov games. The main contribution the proposal of a general multiagent Inverse RL framework, MAGAIL that bridges the gap between existing multiagent RL methods and implicit generative models, such as GANs. MAGAIL is rigorous extension of a previous work on generalized single agent inverse reinforcement learning, GAIL. MAGAIL allows incorporation of prior knowledge into the discriminators, including the presence of cooperative or competitive agents. The paper also proposed a novel multiagent natural policy gradient algorithm that addresses the high variance issue. MAGAIL is evaluated on both cooperative and competitive tasks from the Particle environments, and a cooperative control task where only sub-optimal expert demonstrations are available. Three variants of MAGIAL (centralized, decentralized, zero-sum) are evaluated and outperform several baselines, including behavior cloning and GAIL IRL baseline that operates on each agent separately. This paper is well-written and reasonably accessible (given the mathematical rigor), and the topic is of importance to the machine learning community (both multiagent systems and reinforcement learning). The contributions are nicely laid out, and the structure of the paper leads to a good flow for the reader. The contributions are clear. The experiments are convincing. Minor points: The advantage function in equation (11) is wrongly specified. There is a minus sign missing. In the definition of occupancy measures, it might be better to change T to P to avoid the confusion of transition function.

Reviewer 2



This paper proposes several alternative extensions of GAIL to multi-agent imitation learning settings. The paper includes strong, positive results on a wide range of environments against a suitable selection of baselines. However, insufficient details of the environments is provided to reproduce or fully appreciate the complexity of these environments. If accepted, I would request the authors add these to the appendix and would appreciate details (space permitting) to be discussed in the rebuttal - particularly the state representation. The more pressing point I would like to raise for discussion in the rebuttal is with regard to the MACK algorithm proposed for the generator. The authors make a justified argument for the novelty of the algorithm, but do not thoroughly justify why they used this algorithm instead of an established MARL algorithm (e.g. Multi-Agent DDPG). It seems that MACK is in itself also a contribution to the literature, but it is not directly compared to a suitable existing baseline. Its use also convolutes whether the gains seen by multi-agent GAIL were due to the different priors used or the different RL algorithm used for the generator. Ideally it would be interesting to see an ablation study showing how multi-agent GAIL performs with the different priors proposed in Figure 1 but with TRPO as in the original paper. Additionally, given the choice to use MACK instead of an established multi-agent RL algorithm, a comparison of multi-agent GAIL with MACK to multi-agent GAIL with multi-agent DDPG would provide empirical evidence of the benefit of MACK. For future work, I would suggest the authors revisit an assumption of the paper that the number of experts matches directly the number of agents. However, I don't think this assumption is strictly necessary for some of the multi agent extensions proposed and would be an interesting further empirical study as it enables unique use cases where expert data is limited but could generalise across agents or provide partial insight into how some of the agents should act.

Reviewer 3



This paper proposes a general multi-agent IRL framework or multi-agent imitation learning for general Markov games, which generalizes Generative Adversarial Imitation Learning to multi-agent cases. The learning procedure corresponds to a two-player game between a generator and a discriminator. The generator controls the policies of agents in a distributive fashion, and the discriminator contains a classifier for each agent that is trained to distinguish that agent’s behavior from that of the corresponding expert. Experimental results on a particle environment and a cooperative control task demonstrate that it is able to imitate complex behaviors in high-dimensional environments with both cooperative and adversarial interactions. Unlike most existing work in multi-agent imitation learning where the agents have very specific reward structures, the proposed framework can handle both cooperative and competitive scenarios. However, the most difficult part is the case when the agents are self-interested and the payoffs are general sum. In this case, there may be multiple Nash equilibriums. When the agents learn in a decentralized manner, they may not be converging and jump between equilibriums. It is not clear how such cases are handled in the proposed method. Update: Thanks for the rebuttal. I have read it carefully.

Reviewer 4



This paper proposes a Multi Agent Inverse Reinforcement Learning paradigm by finding connections of multi-agent reinforcement learning algorithms and implicit generative models when working with the occupancy measure. The problem is very important and the solution provided looks interesting. Experiments on cooperative and completive environments establish the effectiveness of the proposed framework. The paper is very well written and easy to follow and read. It could improve by a proof check especially towards the end of the paper. Intuitively, it looks that the generalization over reference [16] is achieved by considering other agents as part of the environment in the training. Is that the case? It’s good to be elaborated and the connections become more clear. The reader may wonder how the method performs without the V_\phi and how the non-stationery of the environment will affect the variance. How much of the success is due to V_\phi and how much is the real contribution of the proposed method. A minor comment is that T is used both for transition function and time horizon. Also, notation-wise D is overloaded. It’s better to change one of them. It’s gonna help the reader grasp the connection to generative adversarial models if the max or min operators in important equations (like 9) are interpreted and their role in the optimization become more clear. The experimental results look convincing. Though it could be more comprehensive to have an empirical ablation study to see how much each part is contributing to the performance. ----------------------- Update: I've read other reviews and the author response. The author response is satisfactory. I'm still in favor of accepting this paper.